# Blockade of CB1 or Activation of CB2 Cannabinoid Receptors Is Differentially Efficacious in the Treatment of the Early Pathological Events in Streptozotocin-Induced Diabetic Rats

**DOI:** 10.3390/ijms24010240

**Published:** 2022-12-23

**Authors:** Dimitris Spyridakos, Niki Mastrodimou, Kiran Vemuri, Thanh C. Ho, Spyros P. Nikas, Alexandros Makriyannis, Kyriaki Thermos

**Affiliations:** 1Department of Pharmacology, School of Medicine, University of Crete, 71003 Heraklion, Greece; 2Center for Drug Discovery, Departments of Chemistry and Chemical Biology and Pharmaceutical Sciences, Northeastern University, Boston, MA 02115, USA

**Keywords:** early stage diabetic retinopathy, endocannabinoid system, cannabinoid receptors, neuroprotection, eye drops, nitrative stress, neuroinflammation, neurodegeneration, vascular leakage

## Abstract

Oxidative stress, neurodegeneration, neuroinflammation, and vascular leakage are believed to play a key role in the early stage of diabetic retinopathy (ESDR). The aim of this study was to investigate the blockade of cannabinoid receptor 1 (CB1R) and activation of cannabinoid receptor 2 (CB2R) as putative therapeutics for the treatment of the early toxic events in DR. Diabetic rats [streptozotocin (STZ)-induced] were treated topically (20 μL, 10 mg/mL), once daily for fourteen days (early stage DR model), with SR141716 (CB1R antagonist), AM1710 (CB2R agonist), and the dual treatment SR141716/AM1710. Immunohistochemical-histological, ELISA, and Evans-Blue analyses were performed to assess the neuroprotective and vasculoprotective properties of the pharmacological treatments on diabetes-induced retinal toxicity. Activation of CB2R or blockade of CB1R, as well as the dual treatment, attenuated the nitrative stress induced by diabetes. Both single treatments protected neural elements (e.g., RGC axons) and reduced vascular leakage. AM1710 alone reversed all toxic insults. These findings provide new knowledge regarding the differential efficacies of the cannabinoids, when administered topically, in the treatment of ESDR. Cannabinoid neuroprotection of the diabetic retina in ESDR may prove therapeutic in delaying the development of the advanced stage of the disease.

## 1. Introduction

The global prevalence of Diabetic Retinopathy (DR) constitutes an alarming health problem. Approximately 4.4 million people worldwide are estimated to be diagnosed with DR, of which one million are blind, and 3.3 million have some kind of vision impairment [1]. DR has recently been categorized into two stages, the early and advanced stages (ESDR and ASDR, respectively). Ophthalmoscopic findings, such as no abnormalities, mild non-proliferative DR (NPDR), and moderate NPDR, are evident in the ESDR, whereas severe NPDR, diabetic macular edema (DME) and proliferative diabetic retinopathy (PDR) in ASDR [2].

Diabetes disturbance of the neurovascular unit (NVU) homeostasis is the main cause of the development of ESDR. The term NVU describes the functional interaction and interdependence among neural, glial, and vascular cells in the retina [3]. Hyperglycemia-induced oxidative stress [4], inflammation [5], and downregulation of pro-survival neurotrophic factors [6] lead to retinal cell damage, such as the progressive degeneration of endothelial cells and pericytes, which are responsible for maintaining the blood–retina barrier (BRB) integrity [7]. Neurodegeneration of retinal neurons has been reported in the pathology of ESDR, with substantial evidence suggesting that neurodegeneration precedes vascular abnormalities [8,9]. Therefore, oxidative stress [10], neuroinflammation [11], neurodegeneration [12], and vascular leakage [13] are all essential components in ESDR pathology.

The endocannabinoid system (ECS) is comprised of the endocannabinoids *N*-arachidonoylethanolamine (AEA), 2-arachidonoylglycerol (2-AG), the enzymes that metabolize and synthesize them, and the receptors, cannabinoid receptor 1 (CB1R) and 2 (CB2R). Both receptors belong to the G protein-coupled receptor (GPCR) superfamily consisting of receptors coupled to pertussis toxin-sensitive inhibitory Gi/o proteins that control a variety of intracellular signaling pathways [14]. All members of the ECS are found in the retina [15]. CB1Rs display extensive expression in various neuronal cell populations, as well as in their synaptic connections [16,17]. CB2Rs can be found mainly in macroglia [18] and microglia [19,20], where they play a crucial role in regulating inflammatory responses [21].

Due to its pleiotropic properties, ECS has been studied as a target for the development of neuroprotective and anti-inflammatory strategies for brain and retinal neurodegenerative diseases [amyotrophic lateral sclerosis (ALS), Parkinson’s disease, multiple sclerosis, glaucoma] [22]. In the retina, numerous studies have shown that CB1R agonists mediate neuroprotective actions against ischemic [23] and excitotoxic insults [20,24,25,26].

However, CB1R activation has also been shown to induce oxidative damage, pro-inflammatory responses [27,28] and to enhance cell apoptosis [29] in various animal models of disease. Blockade of CB1R activation was shown to reduce cell death, oxidative stress, inflammation, and vascular damage in diseases arising from other diabetic complications, such as cardiopathy [30] and nephropathy [31,32]. In a rodent model of DR, it was shown that genetic or pharmacological blockade of CB1R reduced oxidative and nitrative stress and prevented retinal endothelial cell death, as determined by the co-localization of TUNEL-positive cells and the endothelial cell marker isolectin B4 [33]. This was also confirmed by in vitro studies, where CB1R antagonists (AM251, SR141716) attenuated oxidative damage and apoptosis in retinal pigment epithelial (RPE) cells exposed to high glucose [34,35]. In addition, SR141716 blocked photoreceptor and bipolar cell degeneration, decreased glial activation, and prevented the development of abnormal vascular complexes in the *N*-methyl-*N*-nitrosourea (MNU) model [36].

CB2R activation has been shown to attenuate inflammation, oxidative stress, and cell death [37,38]. In a mouse model of diabetic nephropathy (DN), AM1241 (CB2R agonist) prevented the development of various pathological aspects of the disease [39]. In the retina, HU-308 (CB2R agonist) decreased photoreceptor degeneration in a mouse model of age-related macular degeneration [40]. This is in agreement with previous findings from our laboratory, where we showed that the CB2R antagonist, AM630, diminished the neuroprotective and anti-inflammatory actions of the synthetic cannabinoid (*R*)-WIN55,212-2 (CB1R/CB2R agonist) in the in vivo retinal model of AMPA excitotoxicity [20].

The current treatments in clinical practice are efficacious in treating DME and neovascularization, thus focusing on the treatment of ASDR [2]. The discovery of new therapeutics for ESDR, agents that are neuroprotective, anti-inflammatory, and vasculoprotective against diabetic insults, is essential in order to curtail the development of DME/PDR, the eyesight-threatening pathologies.

In order to study the early pathological events of DR and their treatment, we employed the experimental rat model of STZ-induced DR because rats have (a) many similarities to humans (genetic, anatomical, and physiological), (b) very short reproduction cycle and (c) low cost of breeding.

The basic objective of the present study was to investigate the effects of the selective CB1R antagonist, SR141716, and CB2R agonist, AM1710, administered topically as eye drops to the diabetic retina, in preventing the development of nitrative stress, neuroinflammation, neurodegeneration and vascular damage during ESDR. We also examined whether the dual treatment of the above agents exhibited improved efficacy over single treatments. Our findings suggest that the three different cannabinoid treatments examined in our study protect, in a differential manner, the diabetic retina against the four pathologies of ESDR.

## 2. Results

### 2.1. Effect of STZ-Induced Diabetes on Glucose Levels and the Weight of Animals

Measurement of blood glucose levels was performed at three different time points, one day before, one day after, and fourteen days after STZ administration. No differences were observed in glucose levels in the five experimental groups of animals before STZ administration (Control: *n* = 8, 93 ± 7.05; diabetic untreated: *n* = 8, 93 ± 11.88; AM1710: *n* = 8, 92.25 ± 7.88; SR141716: *n* = 8, 91.75 ± 6.27; AM1710 + SR141716:91.75 ± 6.05, *p* > 0.05 compared to Control (Appendix A). One day post initiation of diabetes by STZ, all diabetic animals (untreated and treated) displayed a marked increase in their blood glucose levels compared to Control (Control: *n* = 8, 92.5 ± 4.34) animals (diabetic untreated: *n* = 8, 497.87 ± 52.14; AM1710: *n* = 8, 494.37 ± 56.51; SR141716: *n* = 8, 496.5 ± 66.48; AM1710 + SR141716: *n* = 8, 495.87 ± 54.95, **** *p* < 0.0001 compared to Control) (Appendix A). The blood glucose levels of all diabetic animals remained elevated fourteen days after STZ administration, compared to Control (Control: *n* = 8, 92.62 ± 6.25) animals (diabetic untreated: *n* = 8, 595.62 ± 7.72; AM1710: *n* = 8, 599.12 ± 1.80; SR141716: *n* = 8, 594.5 ± 6.67; AM1710 + SR141716: *n* = 8, 586.37 ± 11.52, **** *p* < 0.0001 compared to Control). There was no statistically significant (*p* > 0.05) difference between untreated and treated diabetic animals (Appendix A).

The effect of STZ administration on the weight of the experimental animals was measured one day prior to STZ treatment and fourteen days after STZ. There was no significant variation (*p* > 0.05) in the weight of the five different groups of animals before STZ administration (Control: *n* = 8, 259.87 ± 9.47; diabetic untreated: *n* = 8, 257.46 ± 7.78; AM1710: *n* = 8, 261.18 ± 6.08; SR141716: *n* = 8, 258.47 ± 8.48; AM1710 + SR141716: *n* = 8, 257.42 ± 7.64) (Appendix A). Fourteen days after STZ injection, the weight of all diabetic animals was significantly reduced compared to Control (Control: *n* = 8, 259.37 ± 9.10) animals (diabetic untreated: *n* = 8, 235.5 ± 10.67, ** *p* = 0.0067 compared to Control; AM1710: *n* = 8, 236.52 ± 6.32, ** *p* = 0.0052 compared to Control; SR141716: *n* = 8, 237.4 ± 5.93, ** *p* = 0.0023 compared to Control; AM1710 + SR141716: *n* = 8, 235.25 ± 4.02, ** *p* = 0.0045 compared to Control) (Appendix A). No significant difference (*p* > 0.05) in weight was observed among all diabetic groups (Appendix A).

### 2.2. CB2R Activation, CB1R Blockade or Their Combination Rescues Ganglion Cells Axons in the Diabetic Retina

The CB2R agonist, AM1710, and the CB1R antagonist, SR141716, administered either alone or in combination, rescued ganglions cell axons from diabetes-induced damage in the retina, as shown by neurofilament (immunoreactivity (NFL-IR) (Figure 1A,B). Quantification studies showed that diabetic untreated animals exhibited a reduction in the intensity of NFL-IR, about 38% compared to Control (Control: *n* = 15, 100.00 ± 11.46; diabetic untreated: *n* = 11, 62.96 ± 16.73, **** *p* < 0.0001 compared to Control) (Figure 1B). A single treatment with AM1710 reversed NFL-IR in diabetic retinas (AM1710: *n* = 7, 83.66 ± 13.25, ^#^ *p* = 0.0403 compared to diabetic untreated) with no statistical difference from the control; *p* > 0.05 compared to Control), whereas SR141716 restored fully NFL-IR (SR141716: *n* = 6, 94.99 ± 20.09, ^###^ *p* = 0.0008 compared to diabetic untreated, *p* > 0.05 compared to Control) (Figure 1B). There was no statistically significant difference in the efficacy of the two treatments (AM1710, *p* > 0.05 compared to SR141716) (Figure 1B). Combined treatment with both AM1710 and SR141716 also diminished diabetes-induced reduction in NFL-IR intensity (AM1710 + SR141716: *n* = 8, 101.10 ± 13.08, ^####^ *p* < 0.0001 compared to diabetic untreated, *p* > 0.05 compared to Control), but a statistically significant difference was not observed between the combined and single treatments (AM1710 + SR141716: *p* > 0.05 compared to AM1710 or SR141716 alone) (Figure 1B).

Similar results were also obtained by the analysis of NFL thickness (Control: 100.00 ± 11.07; diabetic untreated: 67.71 ± 10.36, **** *p* < 0.0001 compared to Control; AM1710: 97.06 ± 4.88, ^####^ *p* < 0.0001 compared to Diabetic untreated, *p* > 0.05 compared to Control; SR141716: 85.73 ± 18.81, ^#^ *p* = 0.0183 compared to Diabetic untreated, *p* > 0.05 compared to Control, *p* > 0.05 compared to AM1710) (Figure 1C). The combination of AM1710 and SR141716 blocked the diabetes-induced reduction in NFL thickness with the same efficacy as the single cannabinoid treatments (AM1710 + SR141716: 99.80 ± 11.00, ^####^ *p* < 0.0001 compared to Diabetic untreated, *p* > 0.05 compared to Control, *p* > 0.05 compared to AM1710, *p* > 0.05 compared to SR141716) (Figure 1C). CF488A goat anti-mouse IgG (H + L) (1:400) stained the OPL and barely the GCL (Appendix A). Therefore, the staining observed mainly in the OPL in Figure 1A is an artifact of the secondary antibody, as stated in the legend of Figure 1. No stain was observed with the Alexa Fluor 488 goat anti-mouse IgG ((H + L) (1:1000) (Appendix A) or with the CF543 goat anti-rabbit IgG (H + L) (1:1000) (Appendix A).

### 2.3. Single Cannabinoid Treatment Blocks in a Differential Manner the Diabetes-Induced Reduction in Retinal Thickness: Diabetes Had No Effect on Ganglion Cells

Cannabinoid treatment with either the CB2R agonist (AM1710) or the CB1R antagonist (SR141716) restored retinal thickness, as shown by the H&E analysis. (Figure 2A,B). Quantitative analysis of the micrographs revealed a significant diabetes-induced reduction of the total retinal thickness (Control: *n* = 5, 100.00 ± 7.90; diabetic untreated: *n* = 4, 80.11 ± 4.86, **** *p* < 0.0001 compared to Control) (Figure 2B). Treatment with the CB2R agonist, AM1710, partially restored retinal thickness (AM1710: *n* = 5, 91.33 ± 3.07, ^#^ *p* = 0.0271 compared to diabetic untreated, * *p* = 0.0161 compared to Control), while the CB1R antagonist, fully restored retinal thickness, in levels similar to the Control retinas (SR141716: *n* = 5, 96.06 ± 3.50, ^##^ *p* = 0.0019 compared to diabetic untreated, *p* > 0.05 compare to Control). Analysis of the three separate retinal nuclear layers, ONL, INL, and GCL (outer nuclear layer, inner nuclear layer, ganglion cell layer), revealed a statistically significant reduction only in the thickness of the INL (Control: *n* = 5, 100 ± 5.6; diabetic untreated: *n* = 4, 70.57 ± 2.10, **
*p* = 0.0034 compared to Control), while both single cannabinoid treatments were able to fully restore this reduction (AM1710: *n* = 5, 97.50 ± 7.77, ^#^*p* = 0.0244 compared to diabetic untreated, *p* > 0.05 compared to Control; SR141716: *n* = 5, 95.42 ± 6.93, ^#^*p* = 0.0383 compared to diabetic untreated, *p* > 0.05 compared to Control) (Figure 2C). In the same sections, we also counted the number of ganglion cells in the GCL, but we did not observe any statistically significant reduction due to diabetes (Control: *n* = 5, 104.30 ± 17.17; diabetic untreated: *n* = 4, 106.10 ± 12.33, *p* > 0.05 compared to Control) (Figure 2D).

### 2.4. Effect of Single Cannabinoid and Dual Treatment on bNOS Expressing Amacrine Cells in the Diabetic Retina: Differential Neuroprotection Effects

Administration via eyedrops of AM1710 reversed the diabetes-induced loss of bNOS expressing retinal amacrine cells, but SR141716 had no effect (Figure 3A,B). As depicted by the quantitative analysis, diabetes caused a reduction in the number of bNOS expressing retinal amacrine cells, about 27% compared to Control retinas (Control: *n* = 12, 100.00 ± 6.48; diabetic untreated: *n* = 13, 73.47 ± 10.80, **** *p* < 0.0001 compared to Control) (Figure 3B). A single treatment with the CB2R agonist, AM1710, fully restored the number of bNOS expressing amacrine cells to levels similar to Control (AM1710: *n* = 7, 104.90 ± 12.26, ^####^ *p* < 0.0001 compared to diabetic untreated, *p* > 0.05 compared to Control) (Figure 3B). In contrast, a single treatment with the CB1R antagonist, SR141716, did not display any protection (SR141716: *n* = 5, 60.51 ± 11.28, *p* > 0.05 compared to diabetic untreated, **** *p* < 0.0001 compared to Control, ^++++^ *p* < 0.0001 compared to AM1710) (Figure 3B). The dual cannabinoid treatment (AM1710 + SR141716) displayed a similar neuroprotective action as that of the treatment with AM1710 (AM1710 + SR141716: *n* = 8, ^#^ *p* = 0.0192 compared to diabetic untreated, *p* > 0.05 compared to Control, ^+++^ *p* = 0.0003 compared to SR141716, ^+^ *p* = 0.0388 compared to AM1710) (Figure 3B).

### 2.5. CB2R Activation but Not CB1R Blockade Reverses Caspase 3-Dependent Apoptotic Cell Death in the Diabetic Retina

Single cannabinoid treatment with the CB2R agonist reduced diabetes-induced apoptotic cell death in the retina, as shown by cleaved caspase 3 immunoreactivity (cl.caspase 3-IR) (Figure 4A,B). Quantitative analysis of cl. caspase 3-IR revealed an increase in the number of cl. caspase 3 positive cells in the INL of diabetic rats compared to Control (Control: *n* = 5, 100.00 ± 33.71; diabetic untreated: *n* = 5, 217.40 ± 29.84, *** *p* = 0.0004 compared to Control). Application of the CB2R agonist AM1710 via eye drops blocked this effect by keeping the number of cl. caspase 3 positive cells, in levels similar to those of Control retinas (AM1710: *n* = 6, 126.70 ± 42.53, ^##^ *p* = 0.0038 compared to diabetic untreated, *p* > 0.05 compared to Control) (Figure 4B). In contrast, when examining the effect of the CB1R antagonist, there was no effect in cl. caspase 3-IR (SR141716: *n* = 7, 166.60 ± 39.41, *p* > 0.05 compared to diabetic untreated, * *p* = 0.0308 compared to Control, *p* > 0.05 compared to AM1710) (Figure 4B).

### 2.6. CB2R Activation Reverses to Control Levels the Diabetes Induced Macroglia Activation

GFAP stains astrocytes and Müller cells. In this study, GFAP was employed as a marker of Müller cell activation. Activation of CB2Rs by AM1710 and blockade of CB1Rs by SR141716 reduced macroglia activation in the diabetic rat retina, as shown by GFAP IR analysis (Figure 5A,B). Quantification studies revealed a significant increase in GFAP staining in the retina of diabetic animals (Control: *n* = 9, 2.55 ± 0.65; diabetic untreated: *n* = 9, 4.39 ± 0.57, **** *p* < 0.0001 compared to Control) (Figure 6B). Only in the presence of AM1710, but not SR141716, the intensity of GFAP-IR was able to be fully restored to Control levels (AM1710: *n* = 5, 2.46 ± 0.50, ^####^ *p* < 0.0001 compared to diabetic untreated, *p* > 0.05 compared to Control; SR141716: *n* = 6, 4.09 ± 0.62, *p* > 0.05 compared to diabetic untreated, *p* = 0.0003 compared to Control, ^+++^
*p* = 0.0008 compared to AM1710) (Figure 5B). The combined treatment, AM1710, and SR141716, was not effective in reducing GFAP expression in the diabetic retina (AM1710 + SR141716: *n* = 6, 4.61 ± 0.41, *p* > 0.05 compared to diabetic untreated, **** *p* < 0.0001 compared to Control, ^++++^ *p* < 0.0001 compared to AM1710, *p* > 0.05 compared to SR141716) (Figure 5B).

### 2.7. CB2R Activation Reduces the Diabetes-Induced Microglia Activation and TNFα Levels

An antibody against Iba1 [resident macrophage (microglia) marker)] was employed in order to assess the effect of diabetes on microglia activation in the retina and the role of CB2R and CB1R in this process (Figure 6A,B). Quantification studies, based on the characteristic morphology of resident macrophages (see Spyridakos et al. [20]), revealed that diabetes led to an increase in the number of Iba1^+^ cells compared to Control animals (Control: *n* = 9, 100.00 ± 30.32; diabetic untreated: *n* = 12, 256.8 ± 101.1, *** *p* = 0.0004 compared to Control) (Figure 6B). Single treatment, using the CB2R agonist AM1710, fully suppressed the activation of microglia, as seen by the reduced number of reactive microglial cells in the treated retina (AM1710: *n* = 6, 102.30 ± 51.44, ^##^ *p* = 0.0010 compared to diabetic untreated, *p* > 0.05 compared to Control) (Figure 6B). On the other hand, the single treatment with the CB1R antagonist, SR141716, was not able to reduce the number of reactive microglial cells in a statistically significant manner (SR141716: *n* = 6, 183.1 ± 60.25, *p* > 0.05 compared to diabetic untreated, *p* > 0.05 compared to Control, *p* > 0.05 compared to AM1710) (Figure 6B) nor did the dual AM1710/SR141716 treatment (AM1710 + SR141716: *n* = 5, 273.9 ± 101.2, *p* > 0.05 compared to diabetic untreated, ** *p* = 0.0047 compared to Control, *p* > 0.05 compared to SR141716, ^+^ *p* = 0.0106 compared to AM1710) (Figure 6B).

The levels of TNFα in the retina were statistically significantly increased in the diabetic retinas, compared to Control (Control: *n* = 4, 271.3 ± 203.2; Diabetic untreated: *n* = 7, 739.3 ± 235.2, ** *p* = 0.0055 compared to Control). AM1710, reduced the diabetes induced elevated levels of TNFα (AM1710: *n* = 6, 408.00 ± 70.09, ^#^
*p* = 0.0290 compared to Diabetic untreated). The CB1R antagonist AM251 had no effect on TNFα elevated levels (AM251: *n* = 5, 476.4 ± 215.3, *p* > 0.05 compared to diabetic untreated, *p* > 0.05 compared to Control, *p* > 0.05 compared to AM1710) (Figure 6C).

### 2.8. CB2R Activation, CB1R Blockade or Their Combination Reduces Nitrative Damage in Diabetic Retina

Cannabinoid treatment, via eye drops, for two weeks was able to reduce nitrative stress in STZ-treated rats, as observed by NT-IR (Figure 7A,B). Quantitative analysis of the data showed that diabetes induced an increase in the number of NT^+^ cells in the retina of STZ-treated rats, compared to Control animals (Control: *n* = 6, 100.00 ± 81.10; diabetic untreated: *n* = 7, 726.70 ± 73.33, **** *p* < 0.0001 compared to Control) (Figure 7B). Administration of AM1710 caused a significant reduction in the number of NT^+^ cells in the diabetic retina (AM1710: *n* = 5, 287.9 ± 110.5, ^####^ *p* < 0.0001 compared to diabetic untreated animals, * *p* = 0.0360 compared to Control) (Figure 7B). A similar result was observed when employing the CB1R antagonist (SR141716: *n* = 5, 435.9 ± 156.1, ^###^ *p* = 0.0004 compared to diabetic untreated, **** *p* < 0.0001 compared to Control, *p* > 0.05 compared to AM1710) (Figure 7B). The combined treatment with both AM1710 + SR141716 was the most efficacious among the three treatments, managing to reduce NT^+^ cells in diabetic retinas at levels relative to those of Control retinas (AM1710 + SR141716: *n* = 7, 101.1 ± 87.17, ^####^ *p* < 0.0001 compared to diabetic untreated, *p* > 0.05 compared to Control, ^+^ *p* = 0.0243 compared to AM1710, ^++++^ *p* < 0.0001 compared to SR141716 treated animals) (Figure 7B).

Quantitative analysis of the different retinal layers revealed that diabetes induced an increase in NT^+^ cells in RPE [1135.64 ± 245.12, ** *p* < 0.0010 compared to Control (10 ± 24.49)], outer plexiform layer (OPL) [690.92 ± 234.68, ^*^*p* < 0.0161 compared to Control (100 ± 113.94)], INL [653.02 ± 173.59, *** *p* < 0.0006 compared to Control (100 ± 126.07)], GCL [485.42 ± 141.33, * *p* < 0.0258 compared to Control (100 ± 109.54)], but not in ONL (*p* > 0.05 compared to Control) (Figure 7C). AM1710 induced a statistically significant reduction in the number of NT^+^ cells in the previously mentioned layers (RPE: 0.85 ± 0.78, ^#^ *p* < 0.0177; OPL: 196.70 ± 81.43, ^#^ *p* < 0.0498; INL: 135.97 ± 56.89, ^#^ *p* < 0.0237; GCL: 150.79 ± 42.26, ^#^ *p* < 0.0451 compared to diabetic untreated) (Figure 7C). SR141716 reduced in a statistically significant manner only the number of NT^+^ cells in the GCL (RPE: 416.46 ± 353.76, *p* > 0.05; OPL: 204.94 ± 153.05, *p* > 0.05; INL: 351.74 ± 122.43, *p* > 0.05; GCL: 248.87 ± 112.53, ^#^ *p* < 0.0187 compared to Control) (Figure 7C). INL SR141716 also displayed a statistically significant difference with AM1710 (^+^ *p* < 0.0297 compared to AM1710) (Figure 7C). AM1710 + SR141716 also reduced nitrative stress in retinal layers (RPE: 60.70 ± 93.19, ^###^ *p* < 0.0002; OPL: 58.30 ± 106.24, ^##^ *p* < 0.0013; INL: 103.72 ± 106.57, ^##^ *p* < 0.0033; GCL: 76.06 ± 81.55, ^##^ *p* < 0.0037 compared to diabetic untreated) (Figure 7C).

### 2.9. Cannabinoid Treatment Reduces the Diabetes-Induced Vascular Leakage

A colocalization study of CD-31, NT, and DAPI was performed in diabetic untreated animals in order to assess the effect of nitrative stress in endothelial cells in the diabetic retina. The study revealed that NT is co-localized with CD-31 positive vessels in GCL (Figure 8A), suggesting that endothelial cells in the diabetic retina are subjected to nitrative stress processes.

Evans-Blue (EB) levels were measured in the diabetic retinas of animals two weeks post-diabetes induction in order to assess the presence of vascular leakage in the early stage of DR. A marked increase in leakage was observed in the diabetic animals compared to Control animals (Control: *n* = 4, 0.09102 ± 0.01150; diabetic untreated; *n* = 7, 0.3146 ± 0.03621, *** *p* = 0.0007 compared to Control) (Figure 8B). Topical administration of the CB2R agonist, AM1710, reduced vascular leakage in the diabetic retina to normal levels (*n* = 5, 0.1283 ± 0.01682, ^##^*p* = 0.0021 compared to diabetic untreated, *p* > 0.05 compared to Control) (Figure 8B). A similar reduction was also observed by the CB1R antagonist, SR141716 (*n* = 3, 0.1160 ± 0.04494, ^##^ *p* = 0.0048 compared to Diabetic untreated, *p* > 0.05 compared to Control) (Figure 8B).

## 3. Discussion

The findings presented in the current study provide new evidence suggesting that topical administration of AM1710 (CB2R agonist) and SR141716 (CB1R antagonist), alone and as dual treatment displayed, in a differential manner, antioxidant, neuroprotective, anti-inflammatory, and vasculoprotective properties in the ESDR model. SR141716 (Rimonabant) is a widely used potent and selective CB1R antagonist [41], while AM1710 is a potent and selective CB2R agonist with very high functional selectivity for CB2R over CB1R [42]. SR141716 has been employed previously in a study investigating the role of CB1R on the development of vascular inflammation and cell death in a mouse model of DR and human retinal cell line [33]. This article was the first to show that CB1R activation plays an important role in the pathogenesis of DR and that CB1R antagonists may be beneficial in reducing diabetes-induced oxidative/nitrative stress and vasculopathy. To our knowledge, AM1710 has never been investigated in healthy or diseased retinas. Therefore, we chose to examine these two cannabinoids, and not others, in the ESDR model with the hope that the single and dual treatments of these compounds will result in novel findings regarding their beneficial use in the treatment of ESDR.

One of the major pathological components of DR is the progressive degeneration of neuronal cell types in the retina. Studies using various experimental models of DR have detected extensive loss of amacrine cells [43], photoreceptors [44], and retinal ganglion cells [45]. These findings have been substantiated by post-mortem data from diabetic human donors [8,46] or by measuring the thickness of retinal layers (OCT) in subjects [47] and in diabetic rats [8]. Numerous reports have shown the neuroprotective actions of CB2R agonists in the brain and their therapeutic potential in a plethora of neurological disorders [48]. In contrast, fewer studies have examined the role of CB2R activation in the retina.

In the present study, we report for the first time that CB2R activation by the selective agonist AM1710, administered topically as eye drops for two weeks, was able to block the diabetes-induced loss of a subpopulation of amacrine cells [nitric oxide synthetase (NOS) expressing].

Another key pathological feature of DR reported in this study was a marked decrease of NFL immunoreactivity, observed two weeks after diabetes induction, which was confirmed both by the quantification of the intensity and thickness of NFL stain. Similar alterations in the morphology of NFL were also reported in five-week [43] and three-month models [49] of STZ-induced DR. The reduction of NFL thickness has also been reported by OCT measurements in human DR patients [50,51]. AM1710 and SR141716, administered either alone or in combination, rescued ganglion cell axons from diabetes-induced damage in the retina [NFL-IR (thickness & intensity)]. To our knowledge, the current study is the first to indicate deficits in NFL as early as two weeks after the onset of diabetes. A number of human studies have reported a reduction in NFL thickness observed at the very early stage of DR, termed preclinical DR [52], or in diabetic patients with no DR [53]. Collectively, these data imply that changes in NFL thickness represent one of the earliest structural changes at the neuronal level of the diabetic retina, thus forewarning the future development of DR.

Despite the reduction in NFL thickness and intensity, as assessed by NFL IR, we did not observe a statistically significant reduction in the thickness of the GCL. Similarly, no difference was observed in the number of retinal ganglion cells (RGCs) between control and diabetic animals (H&E staining). Martin et al. [54] reported that the number of RGCs, as well as the relative thickness of GCL, remained unaffected two weeks post-STZ-induced diabetes in a murine model, in agreement with our findings. However, a statistically significant reduction was observed ten weeks post-diabetes initiation. These results suggest that NFL may be a useful marker to indirectly extract information about the fate of RGCs in optical neuropathies, including DR. Yi et al. [55] reported that axotomy of the optic nerve leads to progressive loss of ganglion cells (optic nerve crush model). The existence of a large number of RGCs, with no detectable retrograde Fluoro-Gold staining, was reported in a mouse model of glaucoma, suggesting that part of their axons was damaged, but RGCs remained functional and expressed various RGC genes [56]. The authors suggested that despite the degeneration of their distal part, the proximal portion of the axons remained unaffected one-month post-insult, thus contributing to RGCs’ survival [56]. These studies support our data and suggest that treatment of diabetes-induced NFL deficits in the ESDR will affect the viability of ganglion cells in a positive manner.

An increased number of TUNEL^+^ cells in all three nuclear retinal layers (ONL, INL, GCL) was reported as early as two weeks after diabetes induction [57]. AM1710 also provided neuroprotection in the ESDR model as suggested by the reduction of the diabetes-induced increase in cleaved caspase-3^+^ cells in the INL, compared to the diabetic untreated animals, and the abolishment of diabetes-induced reduction in INL thickness. These findings are in agreement with previous studies from our laboratory, where it was shown that the neuroprotective properties of the endocannabinoid 2-AG [25] and the synthetic cannabinoid (*R*)-WIN55,212-2 [20] are in part mediated via CB2R activation in the animal model of AMPA induced retinal excitotoxicity. Further proof regarding the neuroprotective properties of CB2R activation was reported by Imamura et al. [40], who showed that HU-308 (CB2R agonist) reduced degeneration of photoreceptors, both in vivo and in vitro, in an animal model of light-induced age-related macular degeneration.

CB1R blockade was reported to be neuroprotective in retinal disease models, such as DR [33] and MNU-induced retinal degeneration [36]. However, CB1R blockade did not reverse the diabetes-induced loss of bNOS expressing amacrine cells nor reduce the number of cleaved caspase-3^+^ cells in the INL. The inability of the CB1R antagonist to influence the attenuation of bNOS amacrine cells and the increase in the number of cleaved caspase 3^+^ cells may be due to the low density of CB1R expressed in nitric oxide synthetase-expressing amacrine cells that are detected (stained) by the bNOS antibody. Furthermore, the CB1R is located mainly in the GCL and less in the INL, where the cell bodies of amacrine cells are located [58]. However, the density of the CB1R in the different amacrine cells has not been reported. Our data are in agreement with a recent study reporting that the CB1R antagonist AM251 aggravated neurodegeneration in a rat model of light-induced retinal degeneration by increasing mRNA levels of pro-apoptotic markers (Bad, Bax) and activated cleaved caspase 3, as well as mRNA levels of the anti-apoptotic marker, Bcl-2. [59]. In the present study, CB1R expression was also examined in three groups (Control, Diabetic, Diabetic + SR141716). No statistically significant difference in CB1R IR was observed, suggesting that neither diabetes nor the treatment affects the expression of CB1R in the retina.

Retinal inflammation is closely linked to persistent macroglia and resident macrophage activation, which are present at the ESDR [60]. In accordance with these findings, we show that diabetes increased the number of Iba1 ^+^ cells and GFAP expression in Müller cells in the two-week model of DR. AM1710 attenuated the diabetes-induced increase in the number of reactive Iba1^+^ cells as well as the levels of TNFα in diabetic retinas, in a statistically significant manner. While the anti-inflammatory actions of CB2R agonists have been well described, this is the first report of its kind using an in vivo model of DR. CB2R activation has been shown to shift the pro-inflammatory M1 phenotype to the anti-inflammatory M2 by stimulating the release of anti-inflammatory cytokines [61]. We have also shown that CB2Rs are expressed in reactive macrophages in the retina, as assessed by their colocalization with Iba1. The synthetic cannabinoid (*R*)-WIN55,212-2 (CB1R/CB2R agonist) attenuated the AMPA-induced increase in the activation of these cells, acting via CB2Rs [20]. AM1710 also reduced macroglia activation in diabetic retinas, as shown by the reduction of GFAP IR. This result is consistent with data showing an up-regulation of CB2Rs in Müller cells [62], suggesting that CB2R plays a crucial role in the physiology of macroglia. Sun et al. [63] reported that the CB2R agonist JWH015 attenuated GFAP expression in astrocytes, an effect that was abolished with the pre-treatment of the animals with the CB2R antagonist AM630.

The mechanism via which CB2R agonists may mediate their anti-inflammatory actions has been reported in different studies. AM1710 activation of CB2R leads to adenylyl cyclase and protein kinase A (PKA) inhibition and attenuation of cAMP levels via Gi/o signaling, that suppress resident macrophage activation via the modulation of c-Jun N-terminal kinases (JNKs) [64]. CB2R activation also leads to the attenuation of NO/nitrative stress and pro-inflammatory cytokines TNF-a and IL-6 via a process that is dependent on ERK1/2 and Akt [65]. CB1R blockade has been suggested to afford anti-inflammatory properties in models of retinopathy, including DR [33,66]. In our study, blockade of CB1R did not attenuate the number of activated macroglia (Figure 5B) and resident macrophage (Figure 6B) nor TNFα levels (Figure 6C) in diabetic retinas, in contrast to our original hypothesis. In a previous study, SR141716 was shown to block diabetes-induced activation of Müller cells (GFAP IR) and NFκB, localized in the vascular layers of the retina, in a mouse STZ model of DR [33]. However, it has been reported that activation, and not a blockade, of CB1R by anandamide was able to attenuate microglia-mediated neuroinflammation in the brain [67]. Despite the ability of SR141716 to reduce nitrative stress and vascular leakage, it did not affect the inflammatory components. As mentioned earlier, this difference may be due to a lower density of CB1Rs located on glial cells.

In agreement with our observation, Solino et al. [59] reported that the application of the CB1R antagonist, AM251, in a rat model of light-induced retinal degeneration increased macroglia reactivity and the expression of TNFα. Another possible explanation for the inability of CB1R blockade to promote anti-inflammatory actions may be due to the existence of a sexual dimorphism effect. A recent study reported that the inflammatory response in the brain is dependent on CB1Rs on resident macrophage cells in male mice but independent of CB1R in female mice [68]. In our study, most of the animals employed were female.

In the model of ESDR, upon onset of diabetes, there was an increase in nitrotyrosine (NT) expression in the retina, a marker of nitrative stress. These findings are in agreement with Hernandez-Ramirez et al. [10], who reported increases in the expression of NO and 3-NT in an STZ-induced DR model seven days post-diabetes initiation. We report that blockade of CB1R and activation of CB2R by topical administration of SR141716 or AM1710, respectively, reduced nitrative stress, as indicated by the attenuation of the diabetes-induced increase in the number of NT+ cells. These findings are in agreement with studies showing CB1R blockade attenuated oxidative damage in various disease models [33,69].

Hyperglycemia-induced oxidative and nitrative stress contribute to microvascular complications of DR by promoting DNA fragmentation and apoptosis of endothelial cells [70]. The reduction of NT levels has been shown to protect retinal micro-vasculature from the nitrative insult [33,71]. In the present study, we observed an increase of NT expression in the diabetic retina (Figure 7A,B) that was reversed by all three treatments. We also showed that NT is localized in endothelial cells, as observed by its co-localization with CD-31 (endothelial cell marker, Figure 8A), suggesting that blood vessels undergo nitrative damage. This finding is confirmed by a previous study showing that NT expression is prominent in the retinal vasculature of diabetic animals [72]. The nitrative stress-induced endothelial cell impairment is a possible explanation for the increased vascular leakage that is also reported in the present study (Figure 8B). We observed an increase in retinal vascular permeability two weeks post-STZ treatment. Studies in diabetic rats reported that increased vascular permeability in the retina was observed as early as six [73] or seven days [74] after STZ administration. AM1710 reduced vascular leakage in diabetic retinas (Figure 8B). In compliance with our results, CB2R activation was shown to reduce the expression of cell adhesion molecules in human retinal endothelial cell cultures and attenuate TNF-α induced retinal vascular permeability in vivo [75,76]. Similar to the actions of AM1710, SR141716 (CB1R antagonist) also blocked the diabetes-induced increase of vascular permeability, suggesting that the blockade of CB1R is efficacious in maintaining the integrity of BRB. Also, vascular protective actions were afforded by SR141716 in a mouse model of retinal degeneration induced by MNU (*N*-methyl-*N*-nitrosourea) [36] and a mouse model of DR [33].

The dual treatment (AM1710 and SR141716) was efficacious in attenuating nitrative stress (Figure 7B,C). However, its efficacy was not different from the single treatments on NFL (Figure 1B,C) or bNOS IR (Figure 3B). Also, it did not afford any statistically significant effect in reducing Müller cell nor resident macrophage activation (Figure 5B and Figure 6B, respectively). Taking into account the anti-inflammatory actions of AM1710 and the inability of SR141716 to affect macroglia and resident macrophage activation, we hypothesized that the dual treatment would mimic single CB2R agonist treatment. Studies in the brain have shown that CB1R and CB2R can form functional heterodimers in neuronal [77], as well as in Iba1^+^ cells. Increased expression of CB1R/CB2R heterodimer has been observed as a response to an inflammatory stimulus [78]. The most interesting part, which seems relevant to our study, is that CB1R/CB2R heterodimers display cross-antagonism in which administration of CB1R antagonists was able to block the effects of CB2R agonists [77,78].

Members of the ECS, endocannabinoids, and CB1/CB2 receptors, found in peripheral organs, are involved in the regulation of energy homeostasis and play an important role in obesity and metabolic disorders [79] Both CB1R antagonists and CB2R agonists modulate glucose metabolism. The synthetic cannabinoid SR141716 (Rimonabant), also employed in this study, was the first selective CB1R antagonist/inverse agonist to be investigated regarding its role in glucose metabolism in patients and animals [80]. Clinical trials [Rimonabant in Obesity (RIO) program] in overweight/obese non-diabetic and type-2 diabetes patients reported that Rimonabant reduced glucose tolerance and insulin resistance [81,82]. Similar results were reported in animal studies employing different CB1R antagonists and various models of obesity [80,83]. The efficacious effects of CB1R antagonists in energy metabolism were negatively affected by their CNS psychiatric adverse effects. In order to bypass this disadvantage, peripheral neutral CB1R antagonists have been developed and were shown to improve glycemic and insulin resistance [84,85]. 2-AG activation was reported to attenuate insulin secretion by pancreatic beta-cells of the mouse via CB2R activation [86] and to improve glucose homeostasis in the rat [87]. All in all, the peripheral endocannabinoid system plays an important role in glucose metabolism. The roles of CB1R antagonist and CB2 agonist in the retina (CNS) and periphery suggest that these members of ECS are prospective putative therapeutics for diabetes and diabetic retinopathy.

In closing, our findings suggest that topical administration of the three cannabinoid treatments, such as eye drops, provides protection to the diabetic retina in a differential manner against the four pathologies of ESDR. The actions of both CB2R activation and CB1R blockade in restoring ganglion cell axons (NFL-IR) in ESDR suggest that both agents may be effective in retarding RGC death. AM1710 is efficacious as an antioxidant, anti-inflammatory, neuroprotective and vasculoprotective agent and, thus, a promising new therapeutic for ESDR. Further advancement of retinal imaging to screen and identify the early events in DR, such as neurodegeneration in diabetic patients, is crucial for selecting neuroprotective drugs and implementing personalized treatments. As our findings clearly implicate the endocannabinoid system, the therapeutic benefits of this class of compounds should also extend to patients with diabetic nephropathy and cardiopathy/stroke since DR has been associated with the development of these diseases [2].

## 4. Materials and Methods

### 4.1. Animals

Adult (2 months old) male and female Sprague Dawley rats, weighing approximately 250–300 g, were used in this study. For the induction of diabetes, mainly female rats were chosen, and the use of male rats was limited when possible due to the high mortality of male subjects after STZ injection. Most studies employ male rats/mice for STZ-induced diabetes. However, taking into consideration the sex differences between males and females [88] in their response to STZ, more investigations using female rats/mice should be proposed. Animals were housed one per cage (Control animals: 2–3 per cage) at a fixed temperature of 22 ± 2 °C and in a stable 12 h light-dark cycle, while food and water were available ad libitum. Inhalation of ascending concentrations of CO_2_ was chosen as the euthanization protocol after the end of the experiments. Animal handling was conducted in accordance with the ARVO Statement for the Use of Animals in Ophthalmic and Vision Research, in compliance with Greek national laws (Animal Act, P.D. 160/91) and the EU Directive for animal experiments (2010/63/EU). All procedures were carried out following reduction and refinement strategies.

### 4.2. Drugs

Streptozotocin (STZ) was obtained from Sigma-Aldrich (Tanfkirchen, Germany). AM1710 (1-Hydroxy-9-methoxy-3-(2-methyloctan-2-yl)-6*H*-benzo[*c*]chromen-6-one) and SR141716(5-(4-Chlorophenyl)-1-(2,4-dichlorophenyl)-4-methyl-*N*-piperidin-1-ylpyrazole -3-carboxamide) were synthesized at the Center for Drug Discovery, Northeastern University (Boston, MA, USA). AM251 (CAS No: 183232-66-8) was purchased from Med Chem Express (MCE, New Jersey, NJ, USA).

### 4.3. Induction of Diabetes and Experimental Design

Prior to the start of all experiments, blood samples were obtained by pricking the lateral tail vein using a sterile needle, and the glucose levels of each animal were measured. Diabetes was induced by a single intraperitoneal (i.p) injection of STZ (70 mg/kg) dissolved in sodium citrate buffer (0.1 M, pH = 4.7), as previously reported [43]. Control animals received a single injection of sodium citrate buffer 0.1 M (vehicle). One day after STZ injections, blood glucose levels were measured to determine the onset of diabetes. Animals were considered diabetic when blood glucose levels were higher than 250 mg/dL. A two-week model of ESDR was employed, according to Hernandez et al. [12], who established the model. The effect of streptozotocin (STZ) on blood glucose levels and weight of animals were analyzed at three-time points (pre-STZ, 1d, and 14d post-STZ treatment) (Appendix A).

Cannabinoid agents were administered as eye drops [20 μL, 10 mg/mL; vehicle: 20 μL DMSO) directly onto the superior corneal surface of each eye using a micropipette once daily for 14 days. Each eye/animal received a different treatment. The 20 μL volume/eye was administered in four different eye drops, ensuring each time that the drop was absorbed. Each animal was restrained by the experimenter with a towel in order to control the movement of the animal’s paws towards the eyes”, Animals were euthanized twenty-four hours after the last eye drop administration, and their tissues were processed. According to the treatment, animals were divided into five different groups, namely, 1. Control, 2. Diabetic, 3. Diabetic animals treated with AM1710 (10 mg/mL in DMSO, 20 μL), 4. Diabetic animals treated with SR141716 (10 mg/mL, in DMSO, 20 μL) and 5. Diabetic animals were treated with the combination of AM1710/SR141716 (10 mg/mL in DMSO, each individually). Animals in control and diabetic non-treated groups received 20 μL of vehicle (DMSO) in each eye as eye drops, as mentioned above, once daily for the same duration as the treatment of the diabetic-treated group.

The eye drop route of administration was chosen because it is more patient-friendly and reaches the retina via the trans-scleral route [89]. The problem with intravitreal injections in humans has to do with the frequency of injections and with the burden on healthcare systems due to the specialized personnel that is needed and the higher cost. Using alternative topical drug delivery, eye drops, for treating the retina is a relevant alternative. However, more investigations are needed in order to determine its usefulness in the treatment of the human retina.

### 4.4. Tissue Preparation

After the verification of euthanization, the eyes were processed for immunohistochemical analysis. Eyes were removed and fixed by immersion in 4% paraformaldehyde (PFA) solution in 0.1 M phosphate buffer (PB) for 45 min at 4 °C. The anterior segment of the eye (containing cornea, lens, aqueous and vitreous humor) was removed, and the eyecup (sclera and retina) was fixed for 1.5 h at 4 °C in addition. Subsequently, the eyecups were cryoprotected by incubation in 30% sucrose solution overnight at 4 °C. Eyecup tissues were frozen in isopentane at −55 to −45 °C, using optimal cutting temperature compound (O.C.T). Vertical sections near the optic nerve head (10 μm thick) were taken using a cryostat (Leica), and slides with tissue sections were stored at −20 °C. For ELISA and Evans Blue assay, after the removal of the eyes, retinas were rapidly isolated and stored at −80 °C until further analysis.

### 4.5. Histology

Cryostat sections were used for histological studies, namely staining with a Hematoxylin and Eosin (H & E) solution, in order to measure the retinal thickness. For this purpose, an H & E staining kit (Abcam, Cambridge, UK, ab245880) was employed, and the detailed protocol followed was the one recommended in the manual of the company.

### 4.6. Immunohistochemical Studies

For the assessment of the effect of the cannabinoid treatment on retinal neurons, a rabbit polyclonal antibody raised against brain nitric oxide synthase (bNOS; 1:2000, Sigma-Aldrich, Tanfkirchen, Germany, code no. N7280, lot no. 062M4839), a marker of retinal amacrine cells and a mouse monoclonal antibody, raised against neurofilament (NFL; 1:500, Millipore, Darmstadt, Germany, Code no. MAB1615, Lot no. 2736736) a marker of ganglion cell axons were employed. The neuroprotective effects of the cannabinoid treatment were also evaluated using a rabbit polyclonal antibody against nitrotyrosine (NT; 1:1000, Millipore, Darmstadt, Germany, Code no. 06-284, Lot 3199176), a marker of nitrative/oxidative stress, and a rabbit monoclonal antibody against cleaved caspase-3 (1:300, Cell Signaling, Danver, MA, USA, Code no., 9661S. Lot no 47), a marker of apoptotic cell death. In order to examine the effect of the cannabinoid treatments on the activation of a) Muller cells and astrocytes, a mouse monoclonal antibody raised against the glial fibrillary acidic protein (GFAP; 1:2000, Sigma-Aldrich, St. Louis, MO, USA, Code no. AB5804, Lot no. 2424641) was employed and b) resident macrophage, a rabbit polyclonal antibody raised against ionized calcium-binding adaptor molecule 1 (Iba1; 1:2500, WAKO Chemicals, Osaka, Japan, Code no. 019-19741). A marker for endothelial cells, cluster of differentiation 31/platelet and endothelial cell adhesion molecule 1 (CD-31/PECAM-1; 1:100, Novus Biologicals, Cambridge, UK, Code no. NB100-64796) was also employed for the visualization of blood vessels. Tissue sections were also treated with the appropriate secondary antibodies. A secondary antibody CF543 goat anti-rabbit IgG (H + L) (1:1000, Biotium, Fremont, CA, USA, code no. 20309, lot no. 12C0213) and a CF488A goat anti-mouse IgG (H + L) (1:400, Biotium, Fremont, CA, code no. 20010, lot no. 13C0619) were used against rabbit and mouse primary antibodies. The CD-31 and NT colocalization study employed the Alexa Fluor 488 goat anti-mouse IgG (H + L) (1:1000, Invitrogen, Waltham, MA, USA, code no. A11029, lot no. 2306576). After completing the incubation of the secondary antibody, all sections were stained with DAPI (1:2000, Sigma-Aldrich, St. Louis, MO, USA) for the visualization of cell nuclei. Tissue sections were also treated alone with the three secondary antibodies, namely CF488A goat anti-mouse IgG (H + L) (1:400), Alexa Fluor 488 goat anti-mouse IgG (H + L) (1:1000), and CF543 goat anti-rabbit IgG (H + L) (1:1000), without the primary antibodies in order to examine their nonspecific staining in the retina.

### 4.7. ELISA Assay

A rat ELISA kit (Abcam, Cambridge, UK, ab100785) was employed for the measurement of the pro-inflammatory cytokine TNF-alpha (TNFα) on retinal samples of Control, diabetic untreated, and diabetic animals treated with either AM1710 or the CB1 inhibitor AM251. An ELISA reader (450 nm, BioRad, Hercules, CA, USA) was used to analyze the duplicates for each sample. TNFα concentration (pg) data were normalized to the total protein concentration (mg) [Nano Drop (2000) (ThermoFisher Scientific, Waltham, MA, USA)].

### 4.8. Evans-Blue Assay for Measuring Blood-Retinal Barrier Permeability

Evans-Blue (EB) dye was diluted in saline 0.9% at a concentration of 30 mg/mL. Twenty-four hours after the last eye-drop administration, animals were anesthetized by intraperitoneal injection of ketamine (100 mg/kg) and xylazine (10 mg/kg). EB solution was injected intravenously via the lateral tail vein at a dosage of 45 mg/kg. Two hours after the administration of EB, blood samples were collected by cardiac puncture, using 19 G needles, and animals were perfused with 0.9% saline at 37 °C in order to remove the excess of dye from the circulation. Additional anesthesia only with ketamine was provided when needed. After perfusion, retinas were isolated and weighed. Blood samples were subsequently centrifuged at 2100 g at 4 °C for 10 min. Plasma samples were diluted in formamide (1/1000) and were prepared for analysis. Retinas were incubated in formamide (300 μL, 72 °C, 18 h), and the extract was centrifuged at 22.000 g (1 h, 4 °C). Retinal extracts and plasma samples were then analyzed as duplicates using a spectrophotometer at 620 nm (maximum absorbance of EB) and 740 nm (minimum absorbance of EB, as previously mentioned [73]. Blood retinal barrier (BRB) leakage was calculated in μg of Evans Blue using the following equation:BRB leakage = (retina EB concentration/plasma EB concentration)/retina wet weight

### 4.9. Microscopy and Quantification Studies

Images from retinal tissue sections were obtained by a fluorescent Leica DMLB microscope (HCX PL Fluotar, 40×/0.70 or 20×/0.50 lens; Leica Micro-systems, Wetzlar, Germany) using a Leica DC 300 F camera. For each immunohistochemical study, magnification was set to either 20× or 40×. All photomicrographs were taken at the center point of the retina, near the optic nerve. Exposure, brightness, contrast, etc., were kept stable until the completion of the quantification analysis. Adobe Photoshop ver. 7.0 software (Adobe Systems, San Jose, CA, USA) was employed to merge images with their representative nuclear staining and crop images in their desirable dimensions. Quantification was performed in the uncropped images, which represent a part of the sagittal section near the optic nerve head (ONH) where the quantification is performed. For each retina (*n* = 1), we employ a total of 6 uncropped images taken from 3 different sections [2 photomicrographs × 3 different sections = 6 images per retina). The mean value of the 6 different uncropped photomicrographs (e.g., cells, thickness, gray value) taken per each sample represents *n* = 1.

For the bNOS-IR study, cells were manually counted across the whole length of tissue sections. For the analysis of NFL-IR, two different protocols were followed. Firstly the mean gray value of NFL-IR was calculated from six different photomicrographs for each sample (two photomicrographs per section, with a total of three sections measured for each sample). Secondly, the thickness of the NFL layer was measured by ImageJ 1.44 software. For each photomicrograph, the NFL thickness was measured at three different points of the photomicrograph width (100, 200, and 300 μm), with the total width of the photomicrograph being 400 μm. Quantification of GFAP-IR was performed according to Anderson et al. [90]. It relies on a scoring system (0–5) to estimate the area of the GFAP positive processes in the tissue. Iba1^+^ reactive cells were identified based on their morphology [20]. The number of reactive Iba1+ cells was manually counted across the whole length of each photomicrograph and expressed as the ratio of their total number to the area measured (expressed in μm^2^). Similarly, manually calculated quantitative analysis was performed for NT^+^ and cleaved caspase-3^+^ cells.

Quantification of the data of the H&E histological study was performed using two nonoverlapping photos from each section taken from the central retina near the optic nerve, with a total number of three sections for each sample. The total thickness of the retina, as well as the thickness of the separate retinal layers (ONL, INL, and GCL), was measured using ImageJ 1.44 software (National Institutes of Health, Bethesda, MD, USA) by drawing straight lines in three different fixed points in each photomicrograph (100, 200, and 300 μm). The approximate width of each photomicrograph was 400 μm. The mean value of each sample was used for the quantitative analysis. The number of ganglion cells in the GCL was also manually counted, beginning 50 μm after the left side and ending 50 μm before the right side of each photomicrograph. The mean of a total of six values for each sample was used for the quantitative analysis.

One possible limitation of this study pertains to the fact that the quantification was performed only on the central part of the retina near the optic nerve head and not the full length of the retina.

### 4.10. Statistical Analysis

Graph Pad Prism 8.0.1 software (San Diego, CA, USA) was employed for the statistical analysis of the data presented in this study. Data are expressed as mean ± S.D and presented in a form that combines bar diagram and scatter dot plot, with every dot representing a different value. Data from all experiments (except ELISA and Evans-Blue assay, where the original values were used) were normalized to the values of Control tissue and expressed as % of Control tissue. One-way analysis of variance (ANOVA) was employed for the statistical analysis of all data, followed by Tukey’s multiple comparison test. A two-way analysis of variance (ANOVA) was employed, followed by Sidak’s multiple comparison test, for the analysis of the different groups across the different retinal layers (ONL, INL, and GCL) in NT and H&E stain. Differences among groups were considered statistically significant when *p* < 0.05. In all statistics, the *n* value represents the number of retinas.

## Figures and Tables

**Figure 1 ijms-24-00240-f001:**
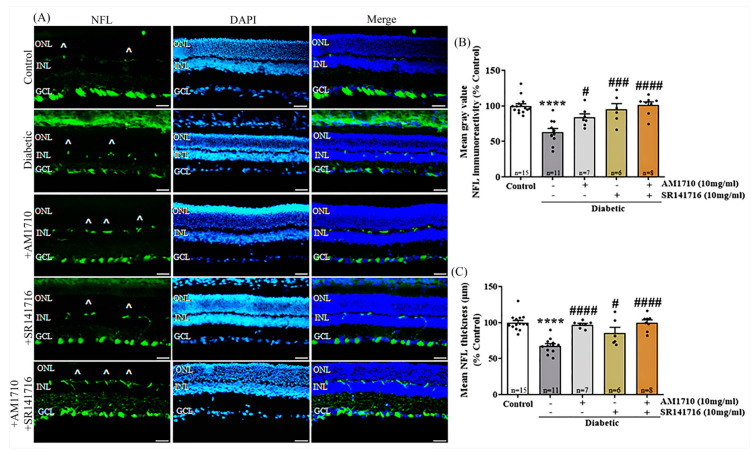
Effect of cannabinoid treatment on ganglion cell axons in the diabetic retina. (**A**) Representative photomicrographs of NFL immunoreactivity (NFL-IR). Artifacts are depicted by the marker (^). Magnification: 20×. Scalebar: 50 μm. (**B**) Quantification studies of NFL-IR: Diabetes attenuated NFL-IR (**** *p* < 0.0001 compared to Control). Both AM1710 and SR141716 reversed the diabetes effect (AM1710; ^#^
*p* = 0.0403 compared to diabetic untreated, *p* > 0.05 compared to Control, and SR141716^; ###^ *p* = 0.0008 compared to diabetic untreated, *p* > 0.05 compared to Control). The dual treatment with AM1710 + SR141716 displayed a similar action by restoring NFL-IR (^####^ *p* < 0.0001 compared to diabetic untreated, *p* > 0.05 compared to Control, *p* > 0.05 compared to AM1710 or SR141716. (**C**) Diabetes induced a significant decrease in NFL thickness (**** *p* < 0.0001 compared to Control). AM1710 and SR141716 reversed the diabetes effect (AM1710; ^####^ *p* < 0.0001 compared to diabetic untreated, *p* > 0.05 compared to Control, and SR141716; ^#^ *p* = 0.0183 compared to diabetic untreated, *p* > 0.05 compared to Control). Dual treatment with AM1710 + SR141716 blocked the diabetes-induced reduction in NFL thickness (^####^ *p* < 0.0001 compared to diabetic untreated, *p* > 0.05 compared to Control *p* > 0.05 compared to AM1710 or SR141716). Data are expressed as mean ± S.D and presented in a form that combines a bar diagram and scatter dot plot, with every dot representing a different value. One-way ANOVA was employed for the statistical analysis of all data, followed by Tukey’s multiple comparison test. Differences were considered statistically significant when *p* < 0.05.

**Figure 2 ijms-24-00240-f002:**
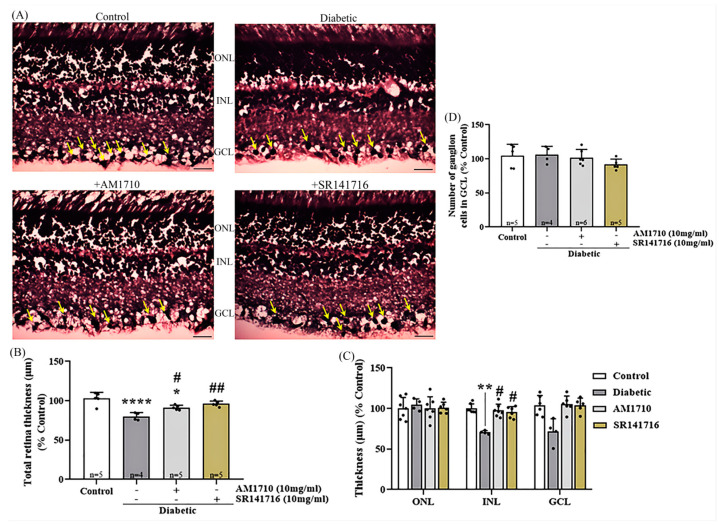
Effect of cannabinoid treatment in the thickness of retinal tissue. (**A**) Representative photomicrographs of H & E stained retinal tissue). Ganglion cells (black) are depicted with yellow arrows. Magnification 20×. Scalebar: 50 μm. (**B**) Quantification analysis of the whole retinal tissue thickness, measured from the retinal pigment epithelium (RPE) to the ganglion cell layer (GCL). Diabetes induced a significant reduction in the total thickness of the retina (**** *p* < 0.0001 compared to Control). Administration of AM1710 partially blocked the reduction in the thickness of the diabetic retina (* *p* = 0.0161 compared to Control, ^#^ *p* = 0.0271 compared to diabetic untreated). SR141716 displayed similar actions (*p* > 0.05 compared to Control, ^##^ *p* = 0.0019 compared to diabetic untreated). (**C**) Quantitative analysis of the thickness of three separate nuclear retinal layers, outer nuclear layer (ONL), inner nuclear layer (INL), and GCL. Diabetes had no effect on the thickness of ONL (*p* > 0.05 compared to Control). INL: The thickness of INL in the diabetic retinas was significantly thinner compared to the retina of Control animals ** *p* = 0.0034 compared to Control). Treatment with either the CB2R agonist (AM1710; ^#^ *p* = 0.0244 compared to Diabetic untreated, *p* > 0.05 compared to Control) or CB1R antagonist (SR141716; ^#^
*p* = 0.0383 compared to diabetic untreated, *p* > 0.05 compared to Control), reversed this reduction of thickness. GCL: The thickness of GCL remained unaffected by diabetes (*p* > 0.05 compared to Control). (**D**) Quantitative analysis of the ganglion cell population in GCL. The number of ganglion cells in GCL remains unaltered by diabetes (*p* > 0.05 compared to Control). Data are expressed as mean ± S.D and presented in a form that combines a bar diagram and scatter dot plot, with every dot representing a different value. One-way ANOVA was employed for the statistical analysis of data on panels (**B**,**D**), followed by Tukey’s multiple comparison test. Two-way ANOVA was employed for the analysis of separate layers’ thickness on panel (**C**), followed by Sidak’s multiple comparison test. Differences were considered statistically significant when *p* < 0.05.

**Figure 3 ijms-24-00240-f003:**
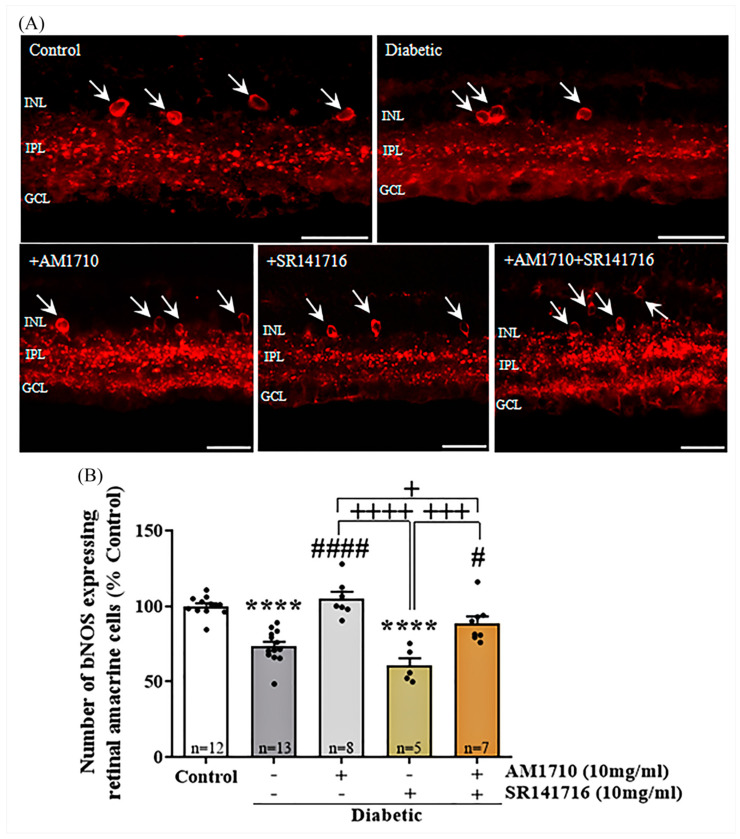
Effect of cannabinoid treatment on the nitric oxide synthetase (bNOS) expressing retinal amacrine cell viability in the diabetic retina. (**A**) Representative photomicrographs of bNOS immunoreactivity (bNOS-IR). White arrows indicate bNOS expressing retinal amacrine cells. Magnification: 40×. Scale bar: 50 μm. (**B**) Quantification of bNOS-IR: Diabetes attenuated in a statistically significant manner the number of bNOS expressing retinal amacrine cells, compared to Control animals (**** *p* < 0.0001 compared to Control). AM1710 reversed the diabetes effect (^####^ *p* < 0.0001 compared to diabetic untreated, *p* > 0.05 compared to Control). SR141716 had no effect (*p* > 0.05 compared to diabetic untreated, **** *p* < 0.0001 compared to Control, ^++++^ *p* < 0.0001 compared to AM1710). The dual treatment, AM1710+ SR141716, blocked diabetes-induced loss of bNOS^+^ cells (^#^ *p* = 0.0192 compared to diabetic untreated, ^+^ *p* = 0.0388 compared to AM1710, ^+++^ *p* = 0.0003 compared to SR141716). Data are expressed as mean ± S.D and presented in a form that combines a bar diagram and scatter dot plot, with every dot representing a different value. One-way ANOVA was employed for the statistical analysis of all data, followed by Tukey’s multiple comparison test. Differences were considered statistically significant when *p* < 0.05.

**Figure 4 ijms-24-00240-f004:**
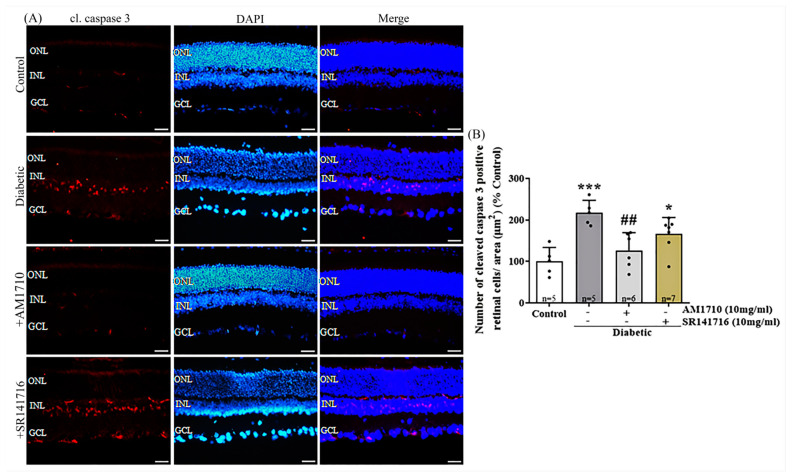
Effect of cannabinoid treatment on cl. caspase-3 dependent apoptosis in the diabetic retina. (**A**) Representative photomicrographs of cleaved caspase-3 immunoreactivity (cl. caspase 3-IR). Magnification: 20×. Scalebar: 50 μm. (**B**) Quantification studies of cl. caspase 3-IR: Diabetes induced an increase in apoptotic caspase 3^+^ cells in the INL of diabetic rats (*** *p* = 0.0004 compared to Control). AM1710 reversed the diabetes effect (^##^ *p* = 0.0038 compared to diabetic untreated, *p* > 0.05 compared to Control). SR141716 (had no effect (*p* > 0.05 compared to diabetic untreated, * *p* = 0.0308 compared to Control, *p* > 0.05 compared to AM1710). Data are expressed as mean ± S.D and presented in a form that combines a bar diagram and scatter dot plot, with every dot representing a different value. One-way ANOVA was employed for the statistical analysis of all data, followed by Tukey’s multiple comparison test. Differences were considered statistically significant when *p* < 0.05.

**Figure 5 ijms-24-00240-f005:**
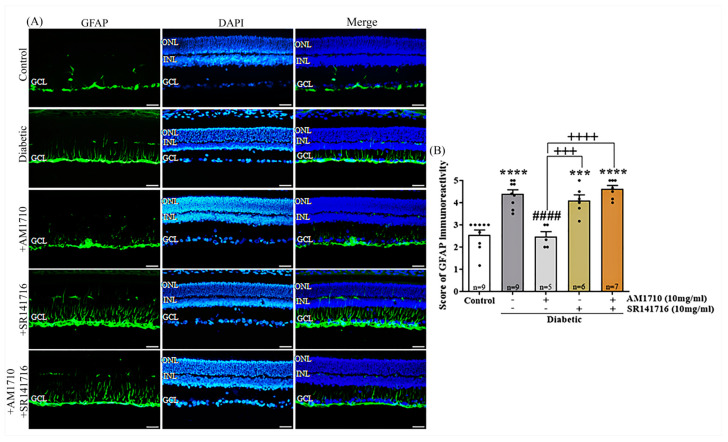
Effect of cannabinoid treatment on Müller cell activation in the diabetic retina. (**A**) Representative photomicrographs of GFAP immunoreactivity (GFAP-IR). Magnification: 20×. Scalebar: 50 μm. (**B**) Quantification studies of GFAP IR. Diabetes increased reactive Müller cells (**** *p* < 0.0001 compared to Control). AM1710 restored GFAP-IR to Control levels (^####^ *p* < 0.0001 compared to diabetic untreated, *p* > 0.05 compared to Control). SR141716 did not affect the diabetes-induced upregulation in GFAP-IR (*p* > 0.05 compared to diabetic untreated, *** *p* = 0.0003 compared to Control, ^+++^ *p* = 0.0008 compared to AM1710). Dual treatment with AM1710+ SR141716 failed to block the diabetes-induced increase in GFAP expression (*p* > 0.05 compared to diabetic untreated, **** *p* < 0.0001 compared to Control, ^++++^ *p* < 0.0001 compared to AM1710). Data are expressed as mean ± S.D and presented in a form that combines a bar diagram and scatter dot plot, with every dot representing a different value. One-way ANOVA was employed for the statistical analysis of all data, followed by Tukey’s multiple comparison test. Differences were considered statistically significant when *p* < 0.05.

**Figure 6 ijms-24-00240-f006:**
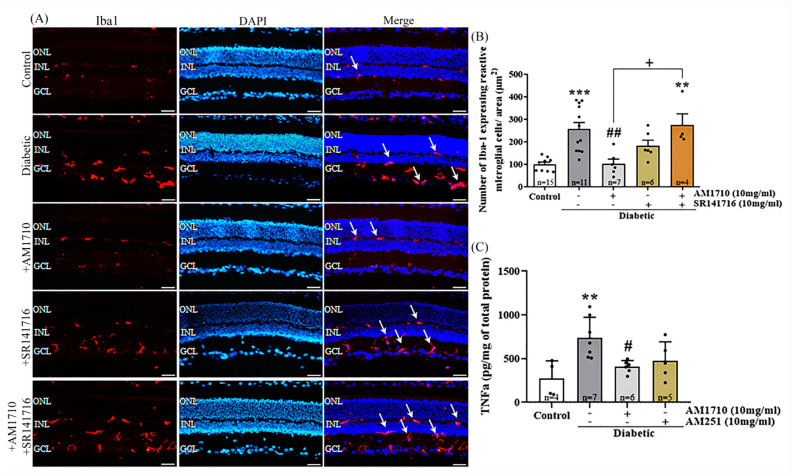
Effect of cannabinoid treatment on macrophage (microglia) activation and pro-inflammatory cytokine (TNFα) release in the diabetic retina. (**A**) Representative photomicrographs of Iba1 immunoreactivity (Iba1-IR). White arrows indicate reactive microglial cells. Magnification: 20×. Scalebar: 50 μm. (**B**) Quantification studies of Iba1-IR: Diabetes increased in a statistically significant manner the number of reactive Iba1^+^ cells (*** *p* = 0.0004 compared to Control). AM1710 reduced Iba1^+^ activation (^##^ *p* = 0.0010 compared to diabetic untreated, *p* > 0.05 compared to Control), while SR141716 had no statistically significant effect (*p* > 0.05 compared to diabetic untreated, *p* > 0.05 compared to Control, *p* > 0.05 compared to AM1710). Dual treatment with AM1710 + SR141716 had no effect on the number of reactive Iba1^+^ cells (*p* > 0.05 compared to diabetic untreated, ** *p* = 0.0047 compared to Control, ^+^ *p* = 0.0106 compared to AM1710, *p* > 0.05 compared to SR141716). (**C**) Quantitative analysis of TNFα levels Diabetes-induced upregulation in TNFα levels in the diabetic retinas (** *p* = 0.0055 compared to Control). AM1710 attenuated this diabetes effect (^#^ *p* = 0.0290 compared to diabetic untreated, *p* > 0.05 compared to Control). No statistically significant effect on TNFα levels was observed by the SR141716 (*p* > 0.05 compared to diabetic untreated, *p* > 0.05 compared to Control). Data are expressed as mean ± S.D and presented in a form that combines a bar diagram and scatter dot plot, with every dot representing a different value. One-way ANOVA was employed for the statistical analysis of all data, followed by Tukey’s multiple comparison test. Differences were considered statistically significant when *p* < 0.05.

**Figure 7 ijms-24-00240-f007:**
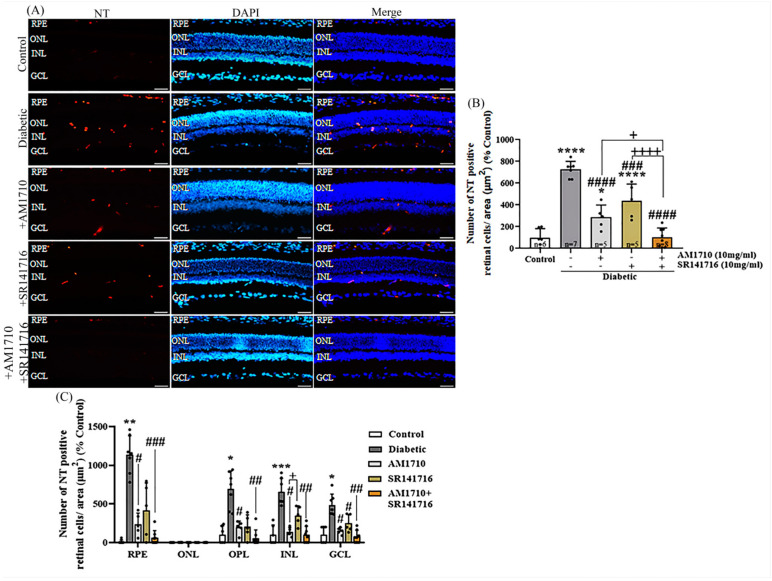
Effect of cannabinoid treatment on nitrative stress induced by diabetes in rat retina. (**A**) Representative photomicrographs of NT immunoreactivity (NT-IR). Magnification: 20×. Scalebar: 50 μm. (**B**) Quantification studies of NT-IR: Diabetes significantly increased the number of NT^+^ (**** *p* < 0.0001 compared to Control). eAM1710 reduced the number of NT^+^ cells in the diabetic retina (^####^ *p* < 0.0001 compared to diabetic untreated animals, * *p* = 0.0360 compared to Control), and SR141716 had a similar effect (^###^ *p* = 0.0004 compared to diabetic untreated, **** *p* < 0.0001 compared to Control, *p* > 0.05 compared to AM1710). The dual treatment AM1710 + SR141716 reversed to Control levels the diabetes-induced increase in nitrative stress (^####^ *p* < 0.0001 compared to diabetic untreated, *p* > 0.05 compared to Control, ^+^ *p* = 0.0243 compared to AM1710, ^++++^ *p* = 0.0001 compared to SR14176). (**C**) Quantification studies of NT-IR in separate retinal layers. Diabetes increased NT-IR in RPE (** *p* < 0.0010), OPL (* *p* < 0.0161), INL (*** *p* < 0.0006), and GCL (* *p* < 0.0258) compared to Control. AM1710 reduced NT-IR in RPE (^#^ *p* < 0.0177), OPL (^#^ *p* < 0.0498), INL (^#^ *p* < 0.0237), and GCL (^#^ *p* < 0.0451) compared to diabetic untreated. SR141716 displayed a statistically significant effect in reducing NT-IR in GCL (^#^
*p* < 0.0187). The dual treatment fully reversed elevated NT-IR in RPE (^###^ *p* < 0.0002), OPL (^##^ *p* < 0.0013), INL (^##^ *p* < 0.0033), and GCL (^##^ *p* < 0.0037), compared to diabetic untreated. Data are expressed as mean ± S.D and presented in a form that combines a bar diagram and scatter dot plot, with every dot representing a different value. One-way ANOVA was employed for the statistical analysis of data on panel (**B**), followed by Tukey’s multiple comparison test. Two-way ANOVA was employed for the analysis of separate layers’ NT-IR on panel (**C**), followed by Sidak’s multiple comparison test. Statistical significance, *p* < 0.05. Differences were considered statistically significant when *p* < 0.05.

**Figure 8 ijms-24-00240-f008:**
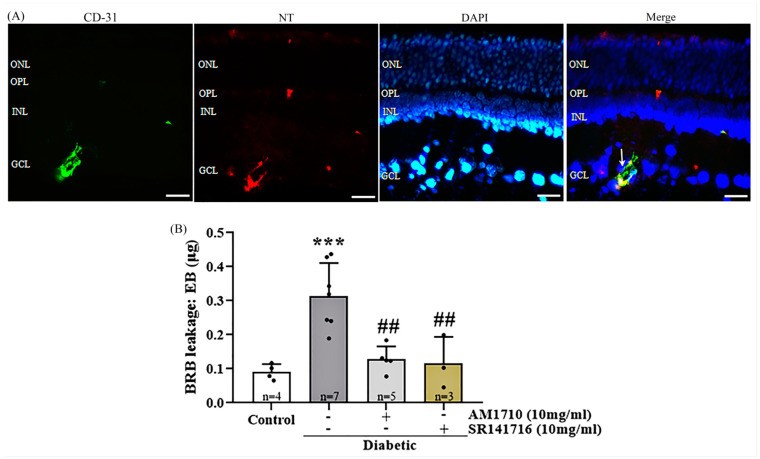
Effect of cannabinoid treatment on diabetes-induced blood–retinal barrier leakage. (**A**) Co-localization of CD-31, NT, and the nuclear marker DAPI. Representative photomicrograph of a diabetic untreated retina, showing that NT is colocalized with CD-31 in blood vessels in the GCL. White arrow marks the co-localization area. Magnification: 20×. Scalebar: 50 μm. (**B**) Quantitative analysis of EB data: Diabetes induced an increase in BRB leakage (*** *p* = 0.0007 compared to Control). Treatment with either AM1710 (^##^ *p* = 0.0021 compared to diabetic untreated, *p* > 0.05 compared to Control) or SR141716 (^##^ *p* = 0.0048 compared to diabetic untreated, *p* > 0.05 compared to Control) blocked this effect. Data are expressed as mean ± S.D and presented in a form that combines a bar diagram and scatter dot plot, with every dot representing a different value. One-way ANOVA was employed for the statistical analysis of all data, followed by Tukey’s multiple comparison test. Differences were considered statistically significant when *p* < 0.05.

## Data Availability

Not applicable.

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
