# Peer review of "Blockade of CB1 or Activation of CB2 Cannabinoid Receptors Is Differentially Efficacious in the Treatment of the Early Pathological Events in Streptozotocin-Induced Diabetic Rats"

_ijms, 2022, doi:10.3390/ijms24010240_

Round 1
Reviewer 1 Report
Spyridakos and colleagues explore the idea that modulating the endocannabinoid system during diabetic retinopathy might exert beneficial effects in treating the pathology. To do so, they use a 2-week type 1 diabetic rat model (STZ) and treated the animals with either a CB1R blocker, an agonist for the CB2R or both at once. They later evaluated the retina of the animals by means of immunohistochemistry to evaluate retina thickness, RGCs axons, gliosis, inflammation (complemented with an ELISA), amacrine cells and oxidation. They complemented this with a vascular permeability assay.
The study is interesting given the limitations in treating diabetic retinopathy. Moreover, the authors served of a topical administration of the drugs, which is extraordinary in the context of retina pathologies. Despite both treatments seem to exert effect to a certain level, these effects were not mirrored in both therapies and the evaluation of the above-mentioned features in the disease retina are rather superficial and lack orthogonal validation.
The manuscript is well-written but abuses of detail in some sections (indicating all quantifications numerically when the figures express this content in the form of bar charts is sometimes redundant). I personally find the discussion a bit “too exhaustive” given that this is a research paper and not a review, therefore I would recommend shortening it. Something important to the discussion is the literature on these treatments and hyperglycemia given the type 1 diabetes strategy used in here. Do these antagonists/agonists modulate glucose metabolism? Are there clinical studies on this? This should be particularly remarked.
The authors must provide better resolution and quality for the figures since it is often hard to distinguish what is a dot and what is a star. Labels are also blurry, and some fonts should be much bigger given the available space.
The authors also cite previous of their work regarding methods, but I believe that this should be avoided in this context given that there is not much diversity in the methods used and they should use this present paper to explain them in detail.
Methodologically, I have 4 major concerns:
1. Despite the treatment is given topically by drops (cornea?), the duration of the treatment is quite long given the 2-week diabetes period. The authors did not mention if the animals are anesthetized during the treatment, and my concern is that if negative, rodents tend to lick very quickly their eyes and the treatment could then influence systemically too, compromising a very important achievement of this work which is to treat locally the retina using drops.
2. The overall methodological approach is rather “old-school” using immunohistochemistry and scoring the fluorescence by-product of the technique. I find important flaws in this as the authors did not provide of isotype negative controls for each antibody used, nor did indicate this in the methods. Also, often the authors indicate that only 3 or 4 pictures per section has been used to quantify the results shown in the figures, and that the quantification has been done manually and not even using semi-automatic methods.
This is unacceptable in 2022 where there is broad availability of free software (including ImageJ scripts) that provide a more unbiased approach. Furthermore, a study that relies entirely in IHC should use the full extent of the retina section and not just few fields captured that, if not randomized properly, can lead to important bias (see for example my comment below regarding Figure 5.
3. Having 2 treatments, and the combination of the 2, is a very important approach and a highlight in this work. I believe the authors secluded these data to supplements since the double treatment was not evaluated for all different readouts. I suggest these data is in main figures given the important and, in the case of histological samples, it should complement the rest of the read-out if there exist additional samples available.
4. I am not sure if I have missed it in the methods, but does the n indicate eye or animal? I would have certain concerns if n = eye and n=2 is from the same animal (biological replicate). Even an n=4 with only 2 animals would be concerning.
With the aim to help the authors to improve the manuscript, I have detailed several (constructive) comments for their consideration:
Line 37: Might be intuitive after NPDR, but it would be good to specify what stands for too here since it is first time mentioned PDR.
Line 55: Typo after reference 15
Line 56: The term macroglia is a bit obsolete as we know that “microglia” are not really glia cells
Line 61: What retina diseases? Please enumerate since it is important to know if the pathologies are vascular, inflammatory, genetic…
Line 70: In reference 34, which cell was prevented to die? Specify.
Line 88 refers only to treatment to advanced DR in DME/PDR, but the STZ model used is more oriented to an early phenotype of the disease.
Figure 1: Why NFL has such strong vascular staining? This is a problem. Isotype negative controls needed.
Figure 2C: Retina thickness should be evaluated in its total length (e.g., spidergram) to reduce bias.
Figure 2D: I honestly cannot see the RGCs in the photographs
Line 149: Typo in a space at “GCL (outer”). Also mention the abbreviation after each term independently and not together at the end of the sentence.
Figure 4: I am a bit concerned about the high levels of cleaved caspase 3 in control animals. Probably need an additional method to analyze this.
Line 225: Typo extra space in “diabetic rats”
234: GFAP stains also astrocytes besides Müller glia, and in the case of Müller glia, is a marker rather related to the phenomenon of gliosis (reactivity). This should be explicitly indicated and taken into consideration.
Figure 5: Retina layers and total length of the diabetic untreated retina look rather thicker than in the control retina photograph. Some photographs lack scale bar too.
Figure 7: RPE seems to express a lot of NT (this is confirmed in Figure 8A). Could the authors perform a quantification distinguishing the different retina layers given that they do not use a cell-specific marker to co-localize signal?
Figure 8B: In which units is EBP expressed? Could the authors show absolute values as well besides a ratio/percentage to control?
Line 419-420: BCL-2 inhibits the execution of the apoptosis. Please verify citation and study carefully.
Line 540: I guess blood glucose was measured from tail vein? Indicate.
Reviewer 2 Report
This study evaluated the putative therapeutics of SR141716 (CB1R antagonist) and AM1710 (CB2R agonist) for the treatment of the early events of diabetic retinopathy (DR) in a Streptozotocin (STZ)-induced diabetic rat model. Some of the results are of interest. Particularly, topical pretreatment of AM1710 via eyedrop attenuated the development of the nitrative stress, neuroinflammation, neurodegeneration and vascular damage. These results are of potential interest for future therapy of early stage of DR. However, there are some issues needed to be addressed before publication.
Major comments
1. The authors need to provide additional data such as Western blot, immunostaining, to verify that activation and/or protein expression of retinal CB1R is inhibited by administration of SR141716 eyedrop, as well as retinal CB2R is activated by AM1710 eyedrop.
2. The authors mainly used female rats for the induction of diabetes. The authors explained the reason is due to the high mortality of male subjects, after STZ injection. However, female rats are less sensitive to STZ (Furman, B. L. (2021). Streptozotocin-induced diabetic models in mice and rats. Current Protocols,1, e78. doi: 10.1002/cpz1.78). Most investigators use only male rats (including Sprague Dawley rats) for STZ-induced diabetes model (using STZ: 45-65mg/kg), while very few studies using female SD rats. The authors shall clarify this issue.
Minor comments:
Title: I think that adding the words “in streptozotocin-induced diabetic rats” at the end of the title would be useful.
Abstract: The authors shall provide the full name for the abbreviation CB1 and CB2 when they were used first time (Line 15)
Introduction
1. It would be better if author add explanation in introduction or discussion section why they choose SR141716 and, AM1710, instead of other CB1R antagonists, and CB2R agonists.
2. Suggest the authors added some comments that they administrate these drugs via eyedrops instead of intravitreal injection
Results
1. Line 103, the authors shall provide the full name for NFL when it was firstly used in the text.
2. Line 106-107, the authors described that “Single treatment with either AM1710 or SR141716, managed to fully restore NFL-IR in diabetic retinas”. It is inappropriate because the NFL-IR in AM178 group is only 83.66% of control.
3. Figures 1, 4-7. From representative photomicrographs of NFL immunoreactivity, either INL or ONL in diabetic group is thicker than control group; The thickness of the two layers in SR141716 group are also thinner than diabetic group. However, the authors showed that the INL decrease in diabetic rats via HE, while both single cannabinoid treatments were able to fully restore this reduction (Line150-153). Authors shall replace them with more representative pictures.
4. Figure 1-8. the authors shall indicate what the error bars are, and what the statistical analysis used in the Figure Legends.
In addition, there are redundant descriptions of the result in the text and Figure legends. For example, in the text, authors described that “Quantification studies showed that diabetic untreated animals, exhibited a reduction in the intensity of NFL-IR, about 38% compared to Control (Control: n=15, 100±11.46; Diabetic untreated: n=11, 62.96±16.73, ****p0.05 compared to Control; SR141716: n=6, 94.99±20.09, ###p=0.0008 compared to diabetic untreated, 109 p>0.05 compared to Control) (Figure 1B), while there was no statistically significant difference in their efficacy (AM1710, p>0.05 compared to SR141716) (Figure 1B). Combined 111 treatment with both AM1710 and SR141716, also diminished the diabetes induced reduction in NFL-IR intensity (AM1710+SR141716: n=8, 101.1±13.08, ####p0.05 compared to Control). In Figure legend, the authors repeated the description that “NFL signal intensity in the GCL was significantly reduced in the diabetic retinas (****p0.05 compared to Control) and the CB1R antagonist, SR141716 (###p=0.0008 compared to diabetic untreated, p>0.05 compared to Control), were able to block the diabetes-induced reduction in the NFL-IR. (C) Diabetes induce a significant decrease in the NFL relative thickness (****p0.05 compared to Control), while the same effect was observed by the use of the CB1R antagonist, SR141716 (#p=0.0183 compared to Diabetic un treated, p>0.05 compared to Control). Suggest that authors concise the description, especially in Figure Legends.
5. It would be better if authors can provide the data of retinal function (for example. ERG examination) after these drugs treatment.
Discussion section
1. Line 352-356. The authors present previous studies about the thickness of retinal layers in patients. Suggest the authors to add some similar studies on diabetic rats in this paragraph.
2. Line 414-420. The authors wrote that “CB1R blockade did not protect bNOS expressing amacrine cells nor reduced the number of cleaved caspase-3 + cells in the INL, indicating the absence of a neuroprotective ability in the ESDR model. The conclusion is inappropriate. This result also could be attributed to the inefficiency concentration and/frequency of CB1R antagonist.
3. Line 456-457. CB1R blockade did not attenuate the number of activated macroglia nor TNFα levels in diabetic retinas. The authors attributed this result to the existence of a sexual dimorphism effect. The reason also could be due to the inefficient concentration and/frequency of CB1R antagonist or the small sample size.
4. Suggest the authors add some comments about the possible mechanisms by which AM1710 reduced macroglia and microglial activation, as well nitrative stress, in diabetic retinas.
Methods
1. Line540-544. Suggest the authors to verify that the blood glucose was fast or non-fast?
2. Line 542. What is the PH value for sodium citrate butter?
3. Line 545-547. Authors wrote that “A two-week model of diabetic retinopathy was employed in this study that was established by Hernandez et al. [13]”. Suggest the authors cite the original article rather than a review paper.
4. Line 547-556. Suggest authors to explain how they determine the concentration, frequency, duration of these eyedrops in either method, result or discussion section.
5. Line 662. for the reference of Iban-Arias et al 2018, the authors shall use the same format as others in the manuscript.
Round 2
Reviewer 1 Report
I appreciate that the authors took into consideration many of my comments and that took their time to address them. The manuscript has improved substantially, but I still have a few comments. I will reply to their response using the numbers (bullet points) to facilitate our correspondence.
1. Ok with me.
2. Nice.
3. I see that the labels are better, but the quality of the images is still insufficient. Please provide high resolution images (no blurriness) for IHC since the authors often refer to findings at a cellular level. Use vectorial files from original microscopy data and export them at high resolution (.TIFF or high resolution .JPG). The figures are too small and because of the resolution, when zooming in, it is hard to tell some features (counting cells). Since this is the driving method (IHC) on the paper, this is important to be addressed.
4. I appreciate this addition. However, I disagree with some points:
Lines 757-760: It is true that using intravitreal injections in a frequent manner (high adherence) can cause inflammation and other of the mentioned complications. However, intravitreal injections are the gold standard in treating DR and the authors cannot state this as the reason to use drops. Proper intravitreal injections (in rodents) do not cause cataract or retinal detachment but require specialized surgical skills. I would propose that the authors rephrase this and emphasize that the problem with the intravitreal injections comes with the frequency, and with the burden to healthcare systems (require specialized personnel and is more expensive). Using alternative topical drug delivery for treating the retina is a relevant alternative to this that need more research efforts.
5a. I am satisfied with this. In fact, showing no changes in blood glucose after treatment is a very elegant manner to depict local vs. systemic pharmacological targeting of the CBS. Please include this data in the manuscript (it can be supplemental).
5b. The authors failed to address the lack of isotype negative controls. Later in this letter they showed some isotype controls, but I cannot find this information in the paper other than indicating in the figure legend “artifacts”. Please address this the methods too. Could the authors specify what they mean by “uncropped”? Do they refer to the full-length retina sagittal section? If this is the case, are these sections comparable to each other among the groups evaluated (e.g., central crossing the ONH, nasal peripheral). Please indicate. Are the authors considering as N for stats the picture (6 photos) or the eye of the subject (1 eye from which the authors used 3 sections?
5c. The authors only used whole-retina section for one single readout, not for the overall analysis of the paper. This does not address my comment. Please express clearly the limitation of this approach.
5d. Alright. So was the n (number of animal) use for all stats?
6. Thank you.
7. Thank you.
8. The argument that many publications use it is not a valid argument. My point is that microglia are mononuclear phagocytes (resident macrophages in the retina and brain) and not glial cells. Macroglia was a term to differentiate the rest of glia (astrocytes, Müller glia) from “microglia” when this fact was not known. I agree that some publications still use this nomenclature, but that doesn’t make it more right and the authors should be more critical with this.
9. Alright.
10. Thanks.
11. I like it, but please add to that rationale that this determined your choice of experimental diabetes in rodents using STZ.
12. I appreciate this. I insist that these data should be include in the manuscript (supplementary figures). When attending to quantification and having positive signal from the isotypes, how is this addressed?
13. I respectfully disagree with the authors. When performing accurate retina sectioning approaching same directionality and region, spidergrams are quite common. Obviously, if sections are not comparable this is not possible (same as any other type of quantification beyond descriptive).
14. I see the arrows; I don’t see the RGCs. Increase the resolution (see my comment above).
15. Thank you.
16. Ok.
17. Ok.
18. Ok.
19. Ok.
20. Nice, I appreciate this.
21. Excellent.
22. Alright.
23. Thanks.
Round 3
Reviewer 1 Report
I had problems to follow the response of the authors. Please refer to lines in the manuscript for future correspondence.
Figure 8: Staining of (mouse monoclonal) CD31 looks the same as the negative isotype control shown in Figure S1. This is not acceptable since the authors are using these results to prove a co-localization of NT. The authors need to demonstrate this using an alternative antibody or experimental approach.
Line 712-736: Could the authors provide specific catalog numbers (not lots) to each primary and secondary antibody and verify that catalog numbers provided are correct? For instance, I could not find the reference provided for GFAP at the Sigma-Aldrich directory.
Line 722: Macroglia should be Müller cell or astrocyte, since the authors do not use any additional marker for Müller cell (f.e. GS, VIM…).
Lines 773, 777: Not relevant, but I have noticed extra typing spaces
Lines 811-821: Statistical N power is still not indicated. The fact that the authors mention before that, for instance, n=1 means 1 retina for microscopy analyses, does not reflect whether the authors will use that n as statistical power, or the N of photographs quantified.
Round 4
Reviewer 1 Report
I understand. Thank you for the explanation and for following the rest of my suggestions.
I think that the piece of data showing a co-localization of NT with the vasculature is important and relevant to diabetic retinopathy, and a highlight of the paper. Before removing the data because of lack of alternative CD31 antibodies and institutional logistics (which I totally understand and I can relate), did the authors tried with a different anti-mouse conjugated with a fluorochrome at a different wave-lenght? Perhaps switching secondaries you can curtail this problem?
